# Dynamical system reconstruction from partial observations using stochastic dynamics

## Abstract

Learning stochastic models of dynamical systems underlying observed data is of
interest in many scientific fields. Here we propose a novel method for this task,
based on the framework of variational autoencoders for dynamical systems. The
method estimates from the data both the system state trajectories and noise time
series. This approach allows to perform multi-step system evolution and supports
a teacher forcing strategy, alleviating limitations of autoencoder-based approaches
for stochastic systems. We demonstrate the performance of the proposed approach
on six test problems, covering simulated and experimental data. We further show
the effects of the teacher forcing interval on the nature of the internal dynamics,
and compare it to the deterministic models with equivalent architecture.

## 1 Introduction

Many scientific fields are concerned with building mathematical models of dynamical systems underlying the observed data. Over the last decade, works using artificial neural networks to achieve
this goal in data-driven fashion have emerged, showing considerable promise (Durstewitz et al.,
2023; Legaard et al., 2023). Compared with the related task of time series prediction, the problem of
dynamical system reconstruction (DSR) is distinguished by the following aspects: focus on the long-
term dynamics of the trained system, an interest in the interpretable structure of the variables and the
latent space, or reasoning and analysis of the learned system by the conceptual and computational
tools from the dynamical system theory.

Many of the influential DSR approaches assume that the underlying dynamics is deterministic (Brun-
ton et al., 2016; Pandarinath et al., 2018; Chen et al., 2018; Hess et al., 2023). Indeed, finding a
deterministic model which can accurately predict future behavior, matches the long-term properties
observed in the data, and correctly generalizes to new conditions can be viewed as the ultimate goal
of DSR. However, reaching such a goal is often infeasible due to the complexity of the underlying
process or due to limited experimental data. Furthermore, reconstructing a deterministic model in
full might not even be desirable if the resulting model is too computationally demanding for the in-
tended purposes. In such cases, relying on a stochastic framework might be a preferred alternative.
Stochastic models include a source of noise in the dynamical equations, in addition to the noisy ob-
servations that are commonly used with deterministic models. This system-level noise can represent
the elements of the system not explicitly modeled by the deterministic part, thus potentially easing
the demands on the deterministic dynamics that needs to be learned. Training of such models, how-
ever, requires different approaches than for deterministic ones. And while various approaches were
explored (Linderman et al., 2017; Duncker et al., 2019; Kramer et al., 2022; Course & Nair, 2023;
Pals et al., 2024, and others discussed in the following section), research into robust and efficient
algorithms with demonstrated ability to learn noise-driven dynamics with various characteristics is
ongoing.

The difficulty of training the system dynamics is influenced by the available observations. In this
work we focus on systems that are only partially observed, that is, where the number of observed
variables is lower than the assumed dimensionality of the system. For such under-observed systems
the time delay embedding (Takens, 1981) in some of its variants (Kraemer et al., 2021) can be used to
reconstruct the trajectory in the state space. Despite some theoretical results on forced and stochastic
systems (Stark et al., 1997), such attempts are in practice mostly limited to autonomous determin-
istic systems. Alternatively, the state trajectory has to be estimated jointly with the training of the

dynamical system. This can be done either directly by optimizing the state variables (Koppe et al., 2019; Kramer et al., 2022), or, more commonly when the training dataset is large, through training an encoder network that maps the observation onto the latent space. Partially observed systems require gathering information from across the temporal dimension, using recurrent neural networks (Pandarinath et al., 2018), temporal convolutional networks (Brenner et al., 2024; Pals et al., 2024), or their combination (García et al., 2022). But these approaches were not yet sufficiently explored, in particular their performance with algorithms used for training stochastic dynamical systems.

In this work, we propose a novel method for DSR of partially observed systems based on stochastic dynamics. To ensure robust training and the ability to capture long-timescale patterns, we introduce a double projection approach, where we map the observations to both the system states and noise time series, on which we train the dynamical model.

Our main contributions are the following:

- First, we propose a novel method for stochastic DSR from minimally observed systems, based on variational autoencoders for dynamical systems and using dual encoding of observation into the state space and noise space.
- Second, we test the method on six test problems, including models of both deterministic chaos and noise-driven dynamics, and experimental data.
- Third, we analyze the nature of the learned dynamics, and investigate the role of the prediction window on it.

## 2 RELATED WORK

A range of methods to train a stochastic dynamical system from the data were developed and explored in recent years. An approach commonly employed for this task - on which we also build in this work - merges probabilistic state space models with the framework of variational autoencoders (VAE, Kingma & Welling, 2014). The key components of the approach are an encoder (or recognition) network mapping the observations into the time series of latent variables, a discrete-time state space model parameterized by a flexible neural network, a decoder (or observation) mapping from the latent space back to the observations, and a training method based on minimizing the evidence lower bound (ELBO). The pioneering works often focused on other applications than DSR, among them are Deep Kalman Filters (Krishnan et al., 2015; 2017) with latent variables being the states of the system, Stochastic Recurrent Networks (Bayer & Osendorfer, 2015) with latent variables being the noise time series, or Variational Recurrent Neural Networks (Chung et al., 2016). These and other related works are reviewed by Girin et al. (2021).

In the explicit context of dynamical system reconstruction, Kramer et al. (2022) used the VAE framework to integrate multimodal data. Brenner et al. (2024) applied it together with a teacher forcing strategy proposed by Hess et al. (2023) for more robust training. Hernandez et al. (2020) trained state-dependent linear networks, reusing the generative model inside the recognition model. Sip et al. (2023) used coupled stochastic models to learn a model of brain network dynamics.

Different methods than dynamical VAEs for training neural network based models of stochastic dynamics exist. Koppe et al. (2019) used the expectation-maximization (EM) method, where the latent states of the dynamical model are optimized directly, as opposed to through an encoder. Kramer et al. (2022) compared the EM method with a VAE-based approach, concluding better performance of EM compared to VAE for smaller problems, but at a cost of limited flexibility. Pals et al. (2024) trained stochastic low-rank recurrent neural networks using the variational sequential Monte Carlo method (Naesseth et al., 2018).

Other works use a different parameterization than neural networks for the stochastic generative model. Linderman et al. (2017) rely on a collection of linear dynamical systems with state-dependent probabilistic switching between them, leading to an interpretable representation. Using Gaussian processes to represent the dynamics of the generative model (Doerr et al., 2018; García et al., 2022) allows to naturally introduce the notion of uncertainty of the dynamics, but it requires careful choice of the kernel and is limited to state spaces of low dimensions.

The works reviewed so far use discrete-time models that can be viewed as an Euler-Maruyama discretization of a continuous stochastic differential equation. The gradients of the loss function

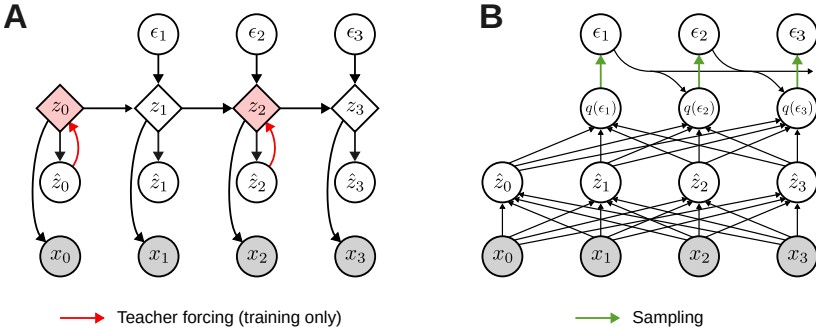

Figure 1: Graphical summary of the DPDSR method. (A) Generation, visualized for teacher forcing interval $\tau = 2$. (B) Encoding. For brevity, we use a shorthand for the posterior distributions $q(\epsilon_t)$ instead of $q(\epsilon_t \mid \boldsymbol{x}, \hat{\boldsymbol{z}}, \boldsymbol{\epsilon}_{1:t-1})$.

necessary for the training are calculated from this discrete formulation, typically using automatic differentiation from modern ML toolboxes: an approach described as discretize-then-differentiate. Continuous-time models proceed in the other direction, differentiate-then-discretize, by formulating differential equations for the gradients, which can then be solved using any appropriate numerical scheme. Among their strengths is the ability to naturally deal with irregularly-spaced observations and better memory scaling properties. Li et al. (2020) generalized the neural ODE framework (Chen et al., 2018) for training continuous-time neural network based stochastic equations. Course & Nair (2023) proposed an efficient way to train the continuous-time stochastic model that avoids solving the differential equation through an ELBO reparameterization.

Dynamical models based on Gaussian processes were also applied for continuous-time approaches. Duncker et al. (2019) used Gaussian processes conditioned on the position of the fixed points and the associated Jacobians, leading to easily interpretable models. Hu et al. (2024) later extended the framework and integrated the ideas of switching linear models (Linderman et al., 2017) by introducing a novel smoothly-switching linear kernel.

## 3 METHODS

### 3.1 DOUBLE PROJECTION DYNAMICAL SYSTEM RECONSTRUCTION

In this study, we consider a dataset of $N$ system observations with dimension $d_x$ over $T$ time steps $\{\boldsymbol{x}_{1:T}^{(i)} \in \mathbb{R}^{T \times d_x}\}_{i=1}^N$. Our aim is to learn an underlying dynamical system

$$z_t = F(z_{t-1}, \epsilon_t),$$
$$x_t = g(z_t) + \Sigma_\eta \eta_t,$$

with system states $z_t \in \mathbb{R}^{d_z}$, system noise $\epsilon_t \in \mathbb{R}^{d_\epsilon}$ of possibly lower dimension $d_\epsilon \leq d_z$, and observation noise $\eta_t \in \mathbb{R}^{d_x}$.

For this goal, we introduce a novel DSR method. We motivate it by the following reasoning: VAE-based approaches are powerful and flexible, but suffer from different drawbacks. For methods where system states $\boldsymbol{z}_{1:T}$ are considered the latent variables (e.g. Deep Kalman Filter (Krishnan et al., 2015; 2017)), the system dynamics are in the loss function expressed by one step transition probability $p(z_t|z_{t-1})$. The system is never let to evolve more than one step from the estimated states, which limits its ability to learn long-term dependencies. On the other hand, for methods where the noise $\boldsymbol{\epsilon}_{1:T}$ are the latent variables (e.g. STORN (Bayer & Osendorfer, 2015)), the absence of known system states prohibits the use of teacher forcing strategies, crucial for training deterministic systems Mikhaeil et al. (2022); Hess et al. (2023).

To circumvent this, we propose a method that uses trained encoders to estimate both the system states and the system noise: Double Projection Dynamical System Reconstruction (DPDSR, Fig. 1). Given one sample of the observed time series $\boldsymbol{x}_{1:T}$, it first estimates the (possibly partial) state space

trajectory $\hat{z}_{1:T}$, and subsequently also the noise time series $\epsilon_{1:T}$. Then, starting from the estimated initial conditions, the system is evolved according to the trained dynamical system and using the estimated noise time series. Every $\tau$ steps, the state of the dynamical system is set to the estimated state $\hat{z}_t$. To calculate the loss function, the match of the generated trajectories to the observations $x_{1:T}$ and the estimated state space trajectory $\hat{z}_{1:T}$ is combined with the Kullback-Leibler (KL) divergence of the latent variables $\epsilon_{1:T}$ from the white noise prior.

In the following text, we write $x = x_{1:T}$ (and analogously for other variables) for readability.

**Generative model**    We consider a generative model of the following form:

$$z_t = \tanh(\, f(z_{t-1}) + B\epsilon_t \,),$$
$$x_t = g(z_t) + \Sigma_\eta \eta_t. \tag{1}$$

The evolution function has the form of a two-layer perceptron with residual connection,

$$f(z_t) = z_t + W_2\sigma(W_1 z_t + b_1) + b_2, \tag{2}$$

with $\sigma(z)$ being the ReLU function. The tanh nonlinearity is added to stabilize the dynamics and training by constraining the states to a finite interval. The observation function is a two-layer perceptron, $g(z_t) = W_2^g\sigma(W_1^g z_t + b_1^g) + b_2^g$. In the examples presented here, we use a one-dimensional noise ($d_\epsilon = 1$) injected into the last dimension only, so that $B = [0, \ldots, 0, \sigma_\epsilon^2]^T \in \mathbb{R}^{d_z \times 1}$. The observation covariance matrix is diagonal and isotropic, $\Sigma_\eta = \sigma_\eta^2 I$.

**Encoder**    The encoding process has two steps. First, from the observed timeseries $x \in \mathbb{R}^{T \times d_x}$ we compute a deterministic estimation of the system state timeseries $\hat{z} \in \mathbb{R}^{T \times d_{\hat{z}}}$. These can be possibly only partial estimation of some dimensions of the states, $d_{\hat{z}} \leq d_z$. We use a WaveNet architecture (van den Oord et al., 2016), which is based on 1D dilated convolutional networks. Unless specified otherwise, we use a single stack of seven dilated convolutional layers (Tab. S1). The output of the last layer is linearly projected to an estimation $\hat{z}$. We are using non-causal layers, although we also train an auxilliary causal stack, which we use for prediction tasks (Sec. A.3.1).

In the second step, we estimate the posterior distribution $q(\epsilon \mid x, \hat{z})$ from the observed time series $x$ and estimated states $\hat{z}$. The noise $\epsilon$ serve as the latent variable in our VAE framework. We use an autoregressive Gaussian posterior, $q(\epsilon_t \mid x, \hat{z}, \epsilon_{1:t-1}) = N(\epsilon_t \mid \mu_t(x, \hat{z}, \epsilon_{1:t-1}), \sigma_t(x, \hat{z}, \epsilon_{1:t-1}))$. The means $\mu_t$ and variances $\sigma_t$ are computed by passing the input timeseries through a WaveNet block with the same architecture as in the first step, followed by an autoregressive LSTM with probabilistic output. This autoregressive form of the posterior distribution serves to increase the expressivity of the encoder of the posterior distribution.

**Training and loss function**    For each time series $x$, the data is first projected to estimate of state $\hat{z}$ and noise $\epsilon$ via the described encoders. If the projection to state space is only partial, the initial conditions are completed by a trainable linear projection to the remained states, $\tilde{z}_0 = [\hat{z}_0; f_{init}(\hat{z}_0)]$. Otherwise, the initial projected state is used, $\tilde{z}_0 = \hat{z}_0$. The system is then evolved according to the generative model (1), using a random sample $\epsilon$ from the posterior distribution. Using teacher forcing, every $\tau$-th step the simulated state is replaced by the estimated state,

$$\tilde{z}_{t+1} = \begin{cases} \tanh(\, f(\tilde{z}_t) + B\epsilon_t \,) & \text{if } t \bmod \tau \neq 0, \\ \tanh(\, f([\hat{z}_t; T_{d_{\hat{z}}:d_z}\tilde{z}_t]) + B\epsilon_t \,) & \text{if } t \bmod \tau = 0. \end{cases} \tag{3}$$

Here, $T_{d_{\hat{z}}:d_z}$ is a truncation matrix selecting the elements from the $d_{\hat{z}}$-th to the last $d_z$-th dimension; that is, the remaining states for which the state was not estimated are left to evolve freely.

For each sample $x$, the loss function is composed from the reconstruction loss of $x$ and $\hat{z}$, and the KL term of the latent noise variables $\epsilon$

$$L = L_x^{\text{rec}} + L_{\hat{z}}^{\text{rec}} + L^{\text{KL}}. \tag{4}$$

The reconstruction loss of the observation is

$$L_x^{\text{rec}} = \mathbb{E}_{\epsilon \sim q(\epsilon \mid x, \hat{z})}[-\log p(x \mid \tilde{z})], \tag{5}$$

where $p(\boldsymbol{x} \mid \tilde{\boldsymbol{z}}) = \prod_t p(x_t \mid g(\tilde{z}_t), \Sigma_\eta)$ using the observation function $g$ and covariance $\Sigma_\eta$ from (1). We treat the estimated states as additional observations with identity mapping, we find that such an approach helps to stabilize the training. Therefore, the reconstruction loss of the estimated partial states is similar to the observation reconstruction loss, apart from replacing the observation operator by the identity function on the estimated states:

$$L_{\hat{z}}^{\mathrm{rec}} = \mathbb{E}_{\boldsymbol{\epsilon} \sim q(\boldsymbol{\epsilon} \mid \boldsymbol{x}, \hat{\boldsymbol{z}})}[-\log p(\hat{\boldsymbol{z}} \mid \tilde{\boldsymbol{z}})], \qquad (6)$$

where $p(\hat{\boldsymbol{z}} \mid \tilde{\boldsymbol{z}}) = \prod_t p(\hat{z}_t \mid T_{1:d_{\hat{z}}} \, \tilde{z}_t, \Sigma_{\hat{z}})$ with $T_{1:d_{\hat{z}}}$ being the truncation matrix selecting the first $d_{\hat{z}}$ states, and the covariance matrix being diagonal and isotropic, $\Sigma_{\hat{z}} = \sigma_{\hat{z}}^2 I$. Finally, the KL term in the loss follows the standard formulation of variational autoencoders,

$$L^{\mathrm{KL}} = D_{\mathrm{KL}}(q(\boldsymbol{\epsilon} \mid \boldsymbol{x}, \hat{\boldsymbol{z}}) \parallel p(\boldsymbol{\epsilon})). \qquad (7)$$

We use a prior of standard normal distribution $p(\boldsymbol{\epsilon}) = N(\boldsymbol{\epsilon} \mid 0, I)$. In addition, two regularization terms are added to the loss. First, to favour sparse observation models, it is L1 regularization on the weights of projection $g$. Second, to encourage desired scale of the state trajectories, it is a regularization term for the scale and position of the estimated states. The loss function is minimized using Adam optimizer. Further details of the architecture and training are given in Sec. A.3.1.

## 3.2 COMPARED METHODS

**Single projection DSR** (SPDSR) is a deterministic variant of DPDSR. It uses the same architecture and teacher forcing training method, but it assumes that there is no noise in the dynamical system. The noise encoder is therefore absent, and the KL divergence term does not appear in the loss function (4). **Generalized teacher forcing** (Hess et al., 2023) is a method for reconstruction of deterministic dynamical system, and has been shown to outperform other deterministic methods. We test two variants: first, using partial forcing (GTF-PF), where only the one observed variable is used as an incomplete teacher signal. And second, using the time-delay embedding (GTF-TD) so that the full state can be forced. We used PECUZAL method for the time delay embedding (Kraemer et al., 2021), and where this failed, we defaulted to a delay embedding with constant offset and predetermined dimension $d = 8$. **Deep Kalman Filter** (DKF) (Krishnan et al., 2015; 2017) is a method for reconstruction of stochastic systems. Using an encoder, it estimates the states of the system, while the one-step stochastic prediction of the generative model forms the basis of the loss function. Our implementation here mirrors the architecture of our proposed method where possible (in the architecture of the encoder and of the generative model), with the main difference being the form of the loss function. **Autoregressive LSTM** (AR-LSTM) (Graves, 2014) uses a standard LSTM network whose probabilistic output is fed back to the network at the next step, thus forming a stochastic dynamical system. During training, the original time series are used to feed the network, or can be replaced by the model generated output following the ideas of scheduled sampling (Bengio et al., 2015). The generative models of all methods were approximately matched in number of parameters (Tab. S2).

## 4 RESULTS

**Example: Double well model** First, we demonstrate the potential of our method and the limits of the alternatives on an example of stochastic dynamics: a noise driven double well model (Fig. 2). The double well dataset (Sec. A.1.3) is generated by numerical integration of a stochastic differential equation of bistable dynamics followed by four layers of exponential smoothing,

$$\dot{z}_1 = -z_1^3 + z_1 + \sigma\eta(t),$$
$$\dot{z}_i = \alpha(z_{i-1} - z_i) \quad \text{for } i \in \{2, 3, 4, 5\},$$

with parameters $\sigma$ scaling the noise amplitude and $\alpha$ being the temporal constant of exponential smoothing. We assume only the last variable $z_5$ is observable. The model has two stable fixed points, $z_i = \pm 1$, $i \in \{1, \ldots, 5\}$.

As shown on Fig. 2C, reconstruction methods based on deterministic dynamics do not perform well on this problem. A good time delay embedding cannot be found for time series generated by stochastic dynamics, and GTF-TD fails to reproduce the bistable nature of the time series. Also partial forcing (GTF-PF) is not sufficient. The embedding-based method using deterministic dynamics

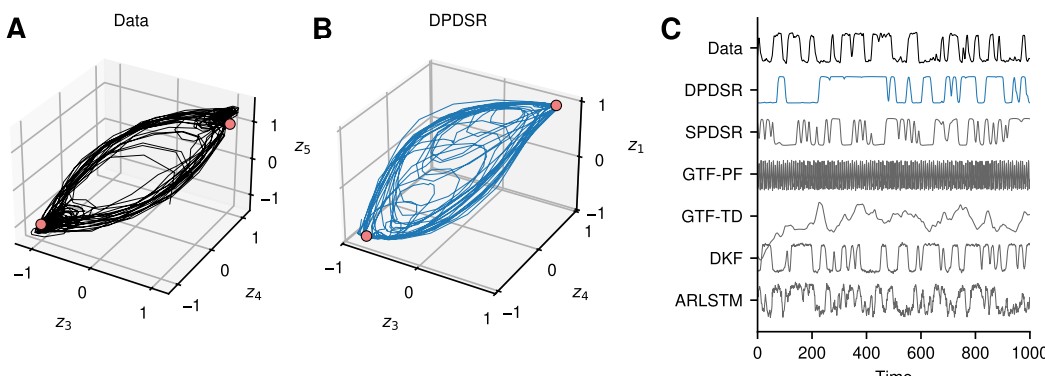

Figure 2: Results for the double well dataset. (A) Trajectory in the original state space of the model. The dots represent the stable fixed points of the model. (B) Simulated trajectory in the state space of the DPDSR model (hand-picked dimensions). (C) Original time series, and time series generated by the trained models. Here, and in all other figures, the time is represented in sample indices and not the original model time.

(SPDSR) performs better and is able to reproduce the bimodal distribution of the data via chaotic dynamics, but a quantitative evaluation in the following section will reveal the inferior match compared to the stochastic method. Among the stochastic methods, both DPDSR and DKF reproduce the bimodal nature of the data well, but DKF suffers from other problems, as we show next. Autoregressive LSTM performs better than the deterministic methods, but worse than embedding-based stochastic methods.

**Quantitative evaluation** For a more detailed evaluation of the DPDSR method, we test it on six datasets (Sec. A.1): 1) Lorenz, the well-known classic example of deterministic chaos (Lorenz, 1963). Even though training models of deterministic chaos is not the main target of the work, it is an important special case for stochastic methods too; 2) Cell cycle, a six-dimensional model of cell division cycle, generated by a deterministic chaotic model (Romond et al., 1999); 3) Double well, the dataset described above; 4) RNN, time series generated by a large recurrent neural network in chaotic regime, of which only one variable is observed. Due to the complexity of the model and minimal observation, we expect that the deterministic dynamics cannot be recovered, and stochastic model would prove to be a better alternative; 5) Neuron, an experimental dataset of somatic voltage of a rat pyramidal neuron driven by random and unknown stimulus current; 6) ECG, an electrocardiogram signal of a healthy adult.

We evaluate the quality of the reconstruction based on three criteria (Sec. A.4). First, the distribution distance $D_d$: the similarity of the spatial distributions of the original and model-generated data. Specifically, we generate long time series with the trained model, collapse the data across time, and compare the distributions using Wasserstein distance. Second, the spectral distance $D_s$: the similarity of the original and model-generated data in the frequency domain. We compute the power spectra of the original and model-generated data, and compare those using the Hellinger distance. And third, the short-term prediction error $\text{PE}_{20}$ of the model-generated trajectories when starting from a state estimated from past data. For the ECG dataset, we also include the distance of interspike intervals $D_{\text{ISI}}$ as an important measure of the reconstruction quality. We summarize these measures in a cumulative score, which we define as a weighted sum of the described measures,

$$S = w_1 D_d + w_2 D_s + w_3 \text{PE}_{20} + w_4 D_{\text{ISI}}.$$

Considering the different nature of the datasets, we set the weights differently for the different datasets in order to reflect the importance of the measures for each specific dataset (Sec. A.4).

We train the models as described in Sec. 3 and Sec. A.3. For each method, we perform a parameter sweep over selected hyperparameters, and for each parameter combination we train the model with four random initializations. We then select the best model as the one with the lowest average score

across initializations. Examples of generated time series are on Fig. S4. The resulting scores for all problems are presented on Fig. 3, and the separate measures on Fig. S5.

We summarize the results in three main points. First, for the datasets generated by low-dimensional deterministic models (Lorenz and Cell cycle), we see a good performance of deterministic models, either based on time delay embedding (GTF-TD) or trained projection (SPDSR). Our stochastic model is, however, competitive in all three measures. DKF performs badly, mainly due to its limited prediction capacity.

Second, for the Double well, RNN, and Neuron datasets, where the time series are dominated by random (or seemingly random) effects on short time scales, the stochastic models DPDSR and DKF show the best performance. The deterministic models (SPDSR and GTF) have to approximate the random transitions through deterministic chaos; the SPDSR projection method does it better than the GTF-TD method based on the time embedding. We note that for these three dataset, the state of the art PECUZAL embedding failed and we used embedding with fixed delay and number of dimension instead. Thus, a worse performance of the time embedding method can be expected. The stochastic AR-LSTM method stands in between the two groups.

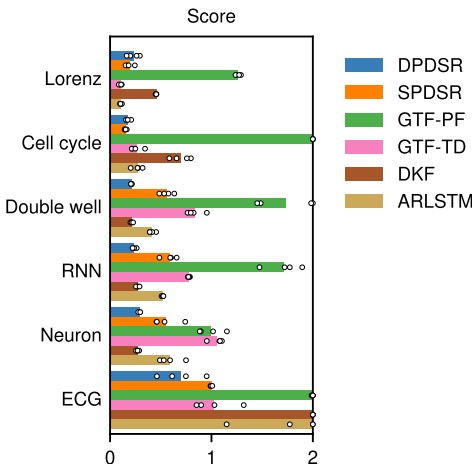

Figure 3: Main results showing the score (lower = better) for all datasets and methods. Each circle represents one of four initialization of the training method, and the bar shows the mean. The measures from which the score is computed are shown on Fig. S5. For visualization purposes, the values are clipped to the upper limit of the shown range.

Third, for the ECG dataset, several method are capable of learning a dynamical system that can generate the periodic ECG signal (Fig. S4). However, closer inspection reveals the variations of the interspike intervals (ISI) in the data (Fig. 4). Unlike our stochastic DPDSR method (Fig. 4B), the deterministic methods do not reproduce the ISI variations (Fig. 4C,D), leading to higher spectral distance and higher overall score. Our stochastic method, however, can exhibit different undesirable behavior. The trained models can "skip a beat" (Fig. 4B); the prevalence of this behavior depends on the random initialization of the training, as demonstrated by the variable results in Fig. 4D.

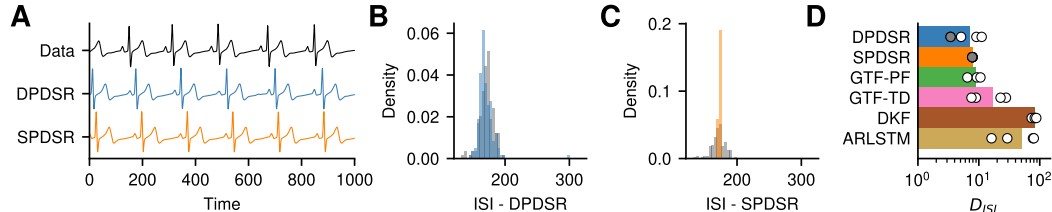

Figure 4: Results for the ECG dataset. (A) Example time series: original, and simulated by the proposed stochastic model (DPDSR) and its deterministic variant (SPDSR). (B) Distribution of the interspike intervals (ISI) in the data (gray) and generated by the DPDSR model. The two distributions mostly overlap, but note the non zero bin near interval 300 indicating skipped beat in the simulated data. (C) Distribution of the ISI in the data (gray) and generated by the deterministic SPDSR model. The model generated data show periodic behavior with all ISIs concentrated on interval 168. (D) Distance between the data and model-generated ISI distributions for all methods. Each circle represent a different training initialization, bar shows their mean. The gray circle represent the model used in panels A-C. The match of ISI distributions in the DPDSR model is strongly dependent on the initialization, which affects the proportion of skipped beats.

**Internal dynamics of the trained models** To better understand the behavior of the proposed DPDSR method, we now analyze the trained models using tools from dynamical systems theory. Specifically, we look at the nature of the attractors of the deterministic part of the trained generative models. For contrast, we compare the results with those from the deterministic SPDSR models. Apart from the absence of the noise, to SPDSR method is identical to the DPDSR in the architecture and training procedure. As such, it provides a useful comparison point for the differences of the stochastic and deterministic methods.

In Fig. 5A we show the maximal Lyapunov exponent of the attractors in the trained models for all six datasets. For each dataset and method, we first detect the attractors by evolving the system from randomly initialized points, and then estimated the maximal Lyapunov exponent (Sec. A.5). The results show that the DPDSR method learns dynamical system exhibiting deterministic chaos for the Lorenz and Cell cycle dataset, both of which are indeed generated by a chaotic system. As shown also on Fig. 2, the model trained on the Double well problem has two stable fixed points (with noise driven transitions between them). The similar RNN problem results in a noise-driven fluctuations around a single fixed point. The variations in the interspike intervals in the ECG model are due to noisy fluctuations around a stable limit cycle.

In contrast, the best performing deterministic models trained with the SPDSR method exhibit chaotic dynamics in all cases. Indeed, in the absence of noise, deterministic chaos is the only option to model the random transitions of the Double well and RNN datasets, or spikes of the Neuron datasets.

Next, we explore the role of the teacher forcing interval $\tau$ on the performance and nature of the dynamics of the trained models. The double well model (Fig. 5B,C) illustrates the phenomena that are to a large extent consistent across all datasets (Fig. S6). In the stochastic DPDSR model, we observe that for smaller intervals ($\tau \leq 40$) the trained models contain chaotic attractors, and they do not rely on the noise. We quantify this by the KL divergence of the estimated posterior distribution of noise from the prior distribution, $\mathrm{KL}(q(\epsilon \mid x) \parallel p(\epsilon))$, averaged across all samples. If the two distributions overlap, the KL divergence is zero, and no information is encoded in the posterior distribution. As the interval $\tau$ increases, the models transition into the regime of noise-driven dynamics with two stable fixed points. This is complemented by the increasing importance of the noise, so that the observed dynamics can be generated by noise-driven fluctuations.

Such behavior is mostly consistent for DPDSR models across datasets, with variations in the position of the transition into the noise-driven regime or the number and nature (fixed point or limit cycle) of the stable attractors. However, the position of the optimal model can differ: it is located in the deterministic chaos regime for Lorenz and Cell cycle, or stochastic regime for Double well,

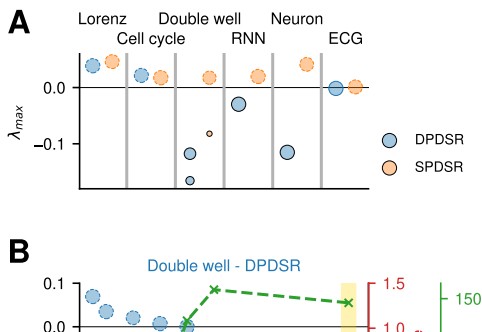

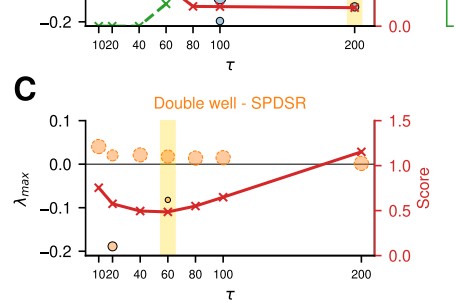

Figure 5: Analysis of the attractors of the trained models. (A) Maximal Lyapunov exponent $\lambda_{\max}$ for the models trained on the six datasets. Each point correspond to one attractor. Dashed circle outline represents a chaotic attractor ($\lambda_{\max} > 0$), colored solid outline represents a limit cycle, and black solid outline a fixed point. Size of the circle corresponds to the size of the basin of attraction. (B) Influence of the teacher forcing interval $\tau$ on the attractors in the DPDSR models. Circles show the maximal Lyapunov exponent as in (A). Solid red line shows the score (lower = better). Dashed green line shows the KL divergence of the estimated posterior distribution of the noise to the prior distribution $\mathrm{KL}_\epsilon = D_{\mathrm{KL}}(q(\epsilon \mid x) \parallel p(\epsilon))$. The yellow band indicates the optimal $\tau$ value for the dataset. (C) As in (B), but for the deterministic SPDSR models.

RNN, Neuron, and ECG problems. In the deterministic SPDSR models, contrastingly, the stochastic regime cannot exist by construction. The models therefore mainly stay in the chaotic regime. The best performing models are found for lower teacher forcing intervals $\tau$, and the score worsens for increasing values. This behavior matches the results reported by Mikhaeil et al. (2022) when training deterministic models using a teacher forcing scheme on the data from Lorenz system, forced Duffing oscillator, and empirical temperature time series.

## 5 DISCUSSION

In this work we have introduced a novel method for reconstructing stochastic dynamical models from the data. The method is based on a double projection approach, where the time series of the observations are projected onto the estimates of the system states and the estimates of the driving noise using trained encoders. The estimated system states are used for teacher forcing, while the driving noise is used as the latent variable in a variational autoencoder framework. We benchmark the method on six test problems, demonstrating its performance for data generated by noise-driven models, deterministic chaotic models, and empirical data (Fig. 3). While other methods, both based on deterministic and stochastic models, can provide equal or even better performance for some of the problems, only our method performs competitively for all tested problems. We then analyze the nature of the attractors in the learned dynamics in the examined problems, and evaluate the role of the teacher forcing interval $\tau$ on it (Fig. 5 and Fig. S6). We identify the existence of two regimes occurring across the test problems: a deterministic regime for lower values of $\tau$ with chaotic attractors, and a noise-driven regime for higher values of $\tau$.

The proposed method has some notable shortcomings and limitations. First, it is the dependency of the behavior of the trained model on the teacher forcing interval. Performing parameter sweeps across a range of values to find the best performing model, while robust, is computationally demanding. As analyzed in Sec. B.1, our results indicate that an adaptive scheme for $\tau$, akin to the scheme of Hess et al. (2023), might not be sufficient to converge to the optimally performing model. The question of optimal strategy for setting the teacher forcing interval thus remains open for future studies.

We have chosen a parameterization of the dynamical system using fully connected neural networks. Such approach offers flexibility in terms of the dynamics it can produce, but limited interpretability, desirable for dynamical system reconstruction. Other works chose different strategies and trade-offs. Symbolic approaches recovering sparse models using predefined function basis (Brunton et al., 2016; Champion et al., 2019) lead to models that are open to pen-and-paper manipulation, but it can be at a cost of reduced performance on empirical datasets (Hess et al., 2023). For specific forms of neural networks with piecewise linear ReLU activation, the fixed points and limit cycles can be found analytically in low dimensions (Schmidt et al., 2021; Brenner et al., 2022; Hess et al., 2023; Pals et al., 2024), greatly simplifying their analysis. Alternatively, Duncker et al. (2019) used Gaussian processes conditioned on the position of fixed points, which are then directly available after training. Switching linear dynamical systems (Linderman et al., 2017; Hu et al., 2024) offer some interpretability by decomposing the state space into regions of linear dynamics.

The ability of a dynamical system reconstruction method to robustly learn dynamics with diverse timescales is of great importance for many problems, and improvements could be made to our proposed methods in this direction. Gated variants of recurrent neural networks, most notably LSTM (Hochreiter & Schmidhuber, 1997) and GRU (Cho et al., 2014), were designed to deal with long-range temporal dependencies. Other approaches with explicit time scale separation were suggested for DSR with deterministic models, often outperforming the traditional architectures. Among the proposed approaches were: using multiple coupled RNNs operating with different temporal resolution (Liu et al., 2022; Farooq et al., 2024), regularization of the parameters of the neural network to introduce multiple time scales (Schmidt et al., 2021), separating time scales using dynamic mode decomposition before applying the DSR algorithm (Bramburger et al., 2020), or using echo state networks with different leak rates of leaky integrator neurons (Tanaka et al., 2022). Applying these ideas to stochastic models could open a way to more robust methods for problems with disparate time scales.

REPRODUCIBILITY STATEMENT

The data and code associated with the study are available in the supplementary material.

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

# A  SUPPLEMENTARY METHODS

## A.1  DATASETS

All datasets were generated or obtained and processed as described below, and all were normalized across time by subtracting the variable mean and dividing by the variable standard deviation.

### A.1.1  LORENZ SYSTEM

The Lorenz system (Lorenz, 1963) is a three-dimensional model exhibiting chaotic behavior, and is among the most used benchmarks in the dynamical system reconstruction field. It is described by three equations,

$$
\begin{aligned}
\dot{x} &= s(y - x), \\
\dot{y} &= rx - y - xz, \\
\dot{z} &= xy - bz,
\end{aligned}
$$

with parameters $s = 10$, $r = 28$, $b = 2.667$. The system was simulated for $T = 10000$ using RK45 method implemented in Scipy library with the default relative and absolute tolerances of $10^{-3}$ and $10^{-6}$ respectively. The simulated time series are exported with sampling period $\Delta t = 0.05$, leaving $200\,000$ time points, divided equally into the train and test time series. Only the first variable $x$ is considered to be observed.

### A.1.2  CELL CYCLE

The model of cell division cycle (Romond et al., 1999) represents an example of deterministic chaos in a six-dimensional state space. It models the evolution of two coupled biochemical oscillators using six differential equations,

$$
\begin{aligned}
\dot{C}_1 &= v_{i1} \frac{K_{im1}}{K_{im1} + M_2} - v_{d1} X_1 \frac{C_1}{K_{d1} + C_1} - k_{d1} C_1, \\
\dot{M}_1 &= V_1 \frac{1 - M_1}{K_1 + (1 - M_1)} - V_2 \frac{M_1}{K_2 + M_1}, \\
\dot{X}_1 &= V_3 \frac{1 - X_1}{K_3 + (1 - X_1)} - V_4 \frac{X_1}{K_4 + X_1}, \\
\dot{C}_2 &= v_{i2} \frac{K_{im2}}{K_{im2} + M_1} - v_{d2} X_2 \frac{C_2}{K_{d2} + C_2} - k_{d2} C_2, \\
\dot{M}_2 &= U_1 \frac{1 - M_2}{H_1 + (1 - M_2)} - U_2 \frac{M_2}{H_2 + M_2}, \\
\dot{X}_2 &= U_3 \frac{1 - X_2}{H_3 + (1 - X_2)} - U_4 \frac{X_2}{H_4 + X_2},
\end{aligned}
$$

with

$$
\begin{aligned}
V_1 &= \frac{C_1}{K_{c1} + C_1} V_{M1}, \quad V_3 = M_1 \cdot V_{M3}, \\
U_1 &= \frac{C_2}{K_{c2} + C_2} U_{M1}, \quad U_3 = M_2 \cdot U_{M3},
\end{aligned}
$$

and $V_{M1} = U_{M1} = 0.3$, $v_{i1} = v_{i2} = 0.05$, $K_{1,2,3,4} = H_{1,2,3,4} = 0.01$, $V_2 = U_2 = 0.15$, $V_{M3} = U_{M3} = 0.1$, $V_4 = U_4 = 0.05$, $K_{c1} = K_{c2} = 0.5$, $v_{d1} = v_{d2} = 0.025$, $K_{d1} = K_{d2} = 0.02$, $k_{d1} = k_{d2} = 0.001$. Following Gilpin (2021), who included the model in a chaotic system collection, we set the value of the bifurcation parameter $K_{im1} = K_{im2} = 0.65$, for which a single chaotic attractor exist in the state space.

The system was simulated for $T = 800000$ using RK45 method implemented in Scipy library with the default relative and absolute tolerances of $10^{-3}$ and $10^{-6}$ respectively, and maximum time step 0.04. The simulated time series are exported with sampling period $\Delta t = 5$, leaving $160\,000$ time points, divided equally into the train and test time series. Only the first variable $C_1$ is considered to be observed.

### A.1.3 DOUBLE WELL

The double well model represents a system with two fixed points and noise driven switches between the two basins of attractions. It is described by a cubic stochastic differential equation with fixed points at -1 and 1, followed by four exponential smoothing equations,

$$\dot{z}_1 = -z_1^3 + z_1 + \sigma\eta(t),$$
$$\dot{z}_i = \alpha(z_{i-1} - z_i) \quad \text{for } i \in \{2, 3, 4, 5\},$$

with $\alpha = 0.4$ and additive Gaussian noise $\eta(t)$ with variance $\sigma^2 = 0.2$. The last variable $z_5$ is considered the observation of the system. The system is simulated with Euler-Maruyama method with $\Delta t = 0.2$ for time $T = 400000$, and then downsampled by a factor of 10, leaving $200\,000$ time points with sampling period 2, equally divided into a train and test set.

### A.1.4 CHAOTIC RNN

The chaotic recurrent neural network model (Sompolinsky et al., 1988) describes the activity of a population of $n$ randomly connected neurons. The dynamics of the synaptic current of the $i$-th neuron is given by

$$\dot{h}_i = -h_i + \sum_{j=1}^{n} J_{ij}\phi(h_j),$$

where $\phi(h) = \tanh(h)$, and the connectivity matrix $\boldsymbol{J}$ contains independent random elements $J_{ij}$ following Gaussian distribution with mean 0 and variance $g/n^2$. We choose the factor $g = 2$, for which the model exhibits chaotic behavior.

We set the number of neurons $n = 1000$, and we solve the system using RK45 method implemented in Scipy library with the default relative and absolute tolerances of $10^{-3}$ and $10^{-6}$ respectively. We solve the system for $T = 100000$ and export with sampling period $\Delta t = 0.5$, leaving $200\,000$ time points, divided equally into the train and test time series. Only the firing rate of the first neuron $\phi(h_1)$ is taken as observed.

### A.1.5 NEURON

The dataset represents the voltage time series of an *in vitro* cortical pyramidal neuron from rat barrel cortex stimulated by a randomly generated fluctuating current (Rauch et al., 2003; Jolivet et al., 2006); the dataset was also used in a spike-timing prediction competition (Jolivet et al., 2008) from which we obtained the data (`https://lcnwww.epfl.ch/gerstner/QuantNeuronMod2007/challenge.html`). In the experiment, the neuron was stimulated by a current generated by an Ornstein-Uhlenbeck process. Although available in the data, here we have assumed that the input current is unknown, and aimed to estimate a stochastic dynamical model of both the neuron and the noise source.

From the available data, we have used only one recording (`00-0`). The signal was sampled at $5\,\text{kHz}$ and contained $34\,000$ data points, we have discarded the initial 600 and final 1200 time points where the stimulus current was not applied. The time series were smoothed with a Gaussian filter with $\sigma = 0.2\,\text{ms}$, normalized over time, and divided equally into the train and test time series of $16\,100$ time points each.

### A.1.6 ELECTROCARDIOGRAM (ECG)

The ECG signal captures the heart's electrical activity over time. We have used human ECG recording from the PPG-DaLiA dataset (Reiss et al., 2019), as preprocessed and used by Hess et al. (2023). They performed smoothing and normalization of the time series, and used the signal of length

$100\,000$ time points at sampling rate $700\,\mathrm{Hz}$ (duration $143\,\mathrm{s}$) for both training and test dataset. We have further downsampled the time series by a factor of 4 ($25\,000$ time points, sampling rate $175\,\mathrm{Hz}$, duration $143\,\mathrm{s}$ for each train and test time series) for reasons of reduced computational costs.

For the sake of consistency with other datasets in this study, we have not used the time embedding provided by Hess et al. (2023), but performed the time embedding ourselves (described below), although using the same embedding method as in the original study. Our approach resulted in 5-dimensional embedding, consistent with the results of Hess et al. (2023).

## A.2 TIME DELAY EMBEDDING

For all time series we first attempted to perform the time delay embedding using the PECUZAL algorithm (Kraemer et al., 2021) using the implementation in DelayEmbedding Julia module. Our settings allowed for possible delay values between 0 and 100 time points, the Theiler window $w$ was set to the first minimum of mutual information of the signal with itself, and we used the economy-mode for L-statistic computation, while all other arguments were set to default. The PECUZAL embedding was successfully achieved for the Lorenz system (dimension $d = 3$), cell cycle model ($d = 5$), and ECG signal ($d = 5$). The PECUZAL embedding failed for the double well system, chaotic RNN, and the neuron recording; in these cases we used a time delay embedding with repeated delays equal to the minimum of mutual information of the signal with itself, and hand-selected embedding dimension $d = 8$.

## A.3 ARCHITECTURE AND TRAINING DETAILS

For all methods, the same procedure was followed to train the models. For each dataset, a parameter sweep was performed over selected hyperparameters (differing across methods), and for each parameter combination, four models were trained with four different random weight initializations. The models were checkpointed during training. The models were evaluated on a separate test set in terms of the score, detailed below. The best parameter combination was chosen by the lowest score averaged across initializations. Where only one model was used for a visualization, the best performing model from the four initializations of the best performing parameter combination was used unless stated otherwise.

### A.3.1 DOUBLE PROJECTION DYNAMICAL SYSTEM RECONSTRUCTION (DPDSR)

The DPDSR method is described in the main text (Sec. 3). Here we describe further details. The regularization term for the observation function is an L1 regularization on the weights of the projection $g$ with coefficient $\alpha_g = 0.3$. The regularization term for the scale and position of the estimated states aims to weakly enforce the desired scale of the state trajectories. It has the form $\alpha_{\hat{z}} D_{\mathrm{KL}}(N(\mu_{\hat{z}}, \mathrm{diag}(\sigma_{\hat{z}}^2) \parallel N(0, I))$, where the mean and variances of the states are calculated across time and samples in a batch, but not features. Denoting the sample in a batch by a superscript, $\mu_{\hat{z}} = \mathbb{E}_{b,t}[\hat{z}_t^b]$ and $\sigma_{\hat{z}}^2 = \mathrm{Var}_{b,t}(\hat{z}_t^b)$. We are using value $\alpha_{\hat{z}} = 0.001$.

Tab. S1 shows the parameters of the encoder model, and Tab. S2 shows the parameters of the generative model. We divide the time series into chunks of $T = 300$, and use batch size 16. To avoid the boundary effects of the convolutional encoders, we evaluate the losses (5-7) on shortened trajectories $\boldsymbol{x}_{1+a:T-a}$ (and equally shortened $\boldsymbol{\epsilon}$, $\tilde{\boldsymbol{z}}$, and $\hat{\boldsymbol{z}}$) instead of the full trajectory $\boldsymbol{x} = \boldsymbol{x}_{1:T}$ with $a = 50$.

For all test problems, the parameter grid exploration was performed over two parameters: teacher forcing interval $\tau \in [1, 10, 20, 40, 60, 80, 100, 200]$, and observation noise variance $\log \sigma_\eta^2 \in [-4, -2, 0]$. The noise variance of the estimated states was set to $\log \sigma_{\hat{z}}^2 = \log \sigma_\eta^2 + 2$ to avoid introducing more free parameters and to reflect the secondary importance of reconstructing the estimated states compared to reconstructing the original observations.

The loss function is minimized using Adam optimizer. The optimizer is ran for 30k batch evaluations, with learning rate starting at 0.001 and reduced by a factor of 0.3 at 10k and 20k steps. The reparameterization trick of variational autoencoders is used to sample from the posterior, with 4 samples used to evaluate the expectations in (5) and (6). We perform gradient clipping with threshold 100. The model is saved every 5000 iterations.

Table S1: Parameters of the encoder models of the DPDSR method.

| | | |
|---|---|---|
| STATE ENCODER | NUMBER OF LAYERS | 7 |
| | KERNEL SIZE | 7 |
| | DILATION | [1,2,4,8,16,32,64] |
| | CHANNELS | 24 |
| NOISE ENCODER | NUMBER OF LAYERS | 7 |
| | KERNEL SIZE | 7 |
| | DILATION | [1,2,4,8,16,32,64] |
| | CHANNELS | 24 |
| | LSTM STATE SIZE | 32 |

Table S2: Parameters of the generative models.

| | | LORENZ | CELL CYCLE ECG | DOUBLE WELL RNN NEURON |
|---|---|---|---|---|
| DPDSR | STATE SIZE | 5 | 8 | 8 |
| | F HIDDEN SIZE | 256 | 256 | 256 |
| | G HIDDEN SIZE | 32 | 32 | 32 |
| | # PARAMETERS | 3015 | 4650 | 4650 |
| SPDSR | STATE SIZE | 5 | 8 | 8 |
| | F HIDDEN SIZE | 256 | 256 | 256 |
| | G HIDDEN SIZE | 32 | 32 | 32 |
| | # PARAMETERS | 3046 | 4681 | 4681 |
| GTF-PF | STATE SIZE | 5 | 8 | 8 |
| | F HIDDEN SIZE | 256 | 256 | 256 |
| | # PARAMETERS | 2826 | 4368 | 4368 |
| GTF-TD | STATE SIZE | 3 | 5 | 8 |
| | F HIDDEN SIZE | 424 | 384 | 256 |
| | # PARAMETERS | 2974 | 4234 | 4368 |
| DKF | STATE SIZE | 5 | 8 | 8 |
| | F HIDDEN SIZE | 256 | 256 | 256 |
| | # PARAMETERS | 2832 | 4377 | 4377 |
| AR-LSTM | STATE SIZE | 24 | 32 | 32 |
| | INITIAL CONDITIONS SIZE | 5 | 8 | 8 |
| | # PARAMETERS | 2642 | 4546 | 4546 |

**Causal encoder**  The DPDSR method uses encoders based on dilated convolutional neural networks. These encoders are non-causal, that is, the estimated states $\hat{z}_t$ at time $t$ are computed from both past and future observations, and therefore $\hat{z}_t$ depends on all elements of $\boldsymbol{x}_{1:T}$. Such approach allows to effectively gather information from the whole provided sample. However, evaluating the predictive performance of such model is difficult, as the initial state would need to be estimated also from future data, invalidating the results. For this reason, we have trained also an auxiliary causal encoder, which uses the same architecture of dilated convolutional networks, but with all connections to future observations set to zero, so that $\hat{z}_t$ depends only on $\boldsymbol{x}_{1:t}$.

The causal encoder is trained in parallel with the main model. In each iteration, the loss and the gradients of the main model are first computed and the update is performed. After that, an update of the causal encoder is performed. Denoting the non-causal encoder as $F$ and causal encoder as $F_c$, the loss function is given by

$$L^{ce} = \|F(\boldsymbol{x}) - F_c(\boldsymbol{x})\|,$$

that is, the causal encoder is trained so its output matches the output of the non-causal encoder. We use Adam optimizer with the same learning rate and same batch sizes as for the main model. The causal encoder is used only for the prediction tasks, while for all other purposes we are using the non-causal variant. The noise posterior distribution is always estimated using the non-causal

encoder, since for the prediction tasks the noise is sampled from the prior and not the posterior distribution, and future information is thus not leaked through it.

### A.3.2 SINGLE PROJECTION DYNAMICAL SYSTEM RECONSTRUCTION (SPDSR)

The architecture of DPDSR allows to consider a special case of a noise-free model by setting $B = 0$ in the generative model (1). In this case, no noise is used in the simulations, and the noise encoder is not used; for this reason we call this special case a Single Projection Dynamical System Reconstruction (SPDSR). In effect, SPDSR is a variant of sparse teacher forcing methods for training deterministic systems, with the teacher signal estimated using the encoder based on convolutional neural networks. We use this variant as a useful comparison point, as it allows us to directly evaluate the effects of the stochastic formulation of DPDSR against a method equivalent in the architecture and training methods.

Indeed, the architecture and training of SPDSR is equal to DPDSR, with the difference that the noise encoder is not employed, and the KL divergence term of the loss function (7) is zero. Same as for DPDSR, the parameter grid exploration was performed over teacher forcing interval $\tau \in [1, 10, 20, 40, 60, 80, 100, 200]$, and observation noise variance $\log \sigma_\eta^2 \in [-4, -2, 0]$. As for the DPDSR method, a secondary causal encoder (Sec. A.3.1) is trained and used for the prediction tasks.

### A.3.3 GENERALIZED TEACHER FORCING (GTF)

Generalized teacher forcing is a training method developed for training deterministic models of chaotic dynamics, and has shown a superior performance on range of problems (Hess et al., 2023). We are using the method to compare our method with state of the art method for deterministic dynamical system reconstruction. Strictly speaking, "generalized teacher forcing" refers only to the training method, but in this paper we use it as a shorthand for both the training method and the specific architecture of the generative model used in the original paper of Hess et al. (2023).

The generative model has the form
$$z_t = A z_{t-1} + W_1 \sigma(W_2 z_{t-1} + h_2) + h_1$$
similar to our parameterization (2) apart from the diagonal matrix $A$, the tanh transformation in our method, and absent noise. The model is trained using a generalized teacher forcing scheme. In every step, the state is replaced by a linear interpolation between the simulated state $\tilde{z}_t$ and the data $z_t$,
$$\hat{z}_t = (1 - \alpha)\tilde{z}_t + \alpha z_t, \tag{8}$$
with coefficient $\alpha \in [0; 1]$. Such approach was shown theoretically to rectify to problem of exploding gradients in learning chaotic dynamics.

We tested two variants of this approach. In the first variant, the full states of the system are estimated using time delay embedding (GTF-TD), and these states are used in the forcing scheme (8). The observations are then equal to the states, $x_t = z_t$. In the second variant we avoid performing time delay embedding and rely on partial forcing instead (GTF-PF). The original time series is used, and only one dimension of the state is forced to the teacher signal in (8), while other variables are left to evolve freely. The observations are then equal to the first state of the system, $x_t = z_{t,1}$.

We used the original implementation in Julia language provided by authors (https://github.com/DurstewitzLab/GTF-shPLRNN).

We followed the example in (Hess et al., 2023) to set the method parameters. We divided the time series into chunks of size $T = 200$ and used batch size 16. The parameters were optimized using RADAM optimizer with 5000 epochs, 50 batches per epoch, and exponential decay schedule with initial and final learning rate $10^{-3}$ and $10^{-6}$ respectively. Every 1000 epochs the model was saved and evaluated, and best performing model was kept. For all test problems, the parameter sweep was performed across the value of teacher forcing parameter $\alpha \in [0.0, 0.1, 0.2, 0.3, 0.4, 0.5]$. The parameters of the generative model are given in Tab. S2.

### A.3.4 DEEP KALMAN FILTER (DKF)

Deep Kalman Filter (Krishnan et al., 2015; 2017) is based upon the framework of variational autoencoders applied to dynamical systems, where the states of the system $z$ are considered to be the

latent variables. The generative model of DKF is

$$z_t = f(z_{t-1}) + \Sigma_\epsilon \epsilon_t,$$
$$x_t = g(z_t) + \Sigma_\eta \eta_t.$$

We use the parameterization of the evolution function

$$f(z_t) = z_t + W_2 \sigma(W_1 z_t + b_1) + b_2$$

and linear observation function $g$. This formulation differs from the one used in the DPDSR method (2) by the absence of the tanh nonlinearity; we have found that this stabilization is not needed for training the DKF models.

During training, for an observation $\boldsymbol{x}$, DKF first projects the observation to a posterior distribution state trajectories $q(\boldsymbol{z} \mid \boldsymbol{x})$. Then a state trajectory is sampled from the posterior, $\boldsymbol{z} \sim q(\boldsymbol{z} \mid \boldsymbol{x})$, and the loss function for the data point is calculated as

$$L(\boldsymbol{x}) = D_{\mathrm{KL}}(q(\boldsymbol{z} \mid \boldsymbol{x}) \parallel p(\boldsymbol{z})) - \mathbb{E}_{\boldsymbol{z} \sim q(\boldsymbol{z}|\boldsymbol{x})}\left[\log p(\boldsymbol{x} \mid \boldsymbol{z})\right].$$

The prior for the system states is

$$p(\boldsymbol{z}) = p(z_0) \prod_{t=1}^{T} p(z_{t+1} \mid z_t),$$

with $p(z_0) = N(z_0 \mid 0, I)$ and $p(z_{t+1} \mid z_t) = N(z_{t+1} \mid f(z_t), \Sigma_\epsilon)$. The probability in the reconstruction loss is

$$p(\boldsymbol{x} \mid \boldsymbol{z}) = \prod_{t=1}^{T} N(x_t \mid g(z_t), \Sigma_\eta).$$

The system and observation noise covariance matrices were assumed to be diagonal and isotropic, $\Sigma_\epsilon = \sigma_\epsilon^2 I$ and $\Sigma_\eta = \sigma_\eta^2 I$. Compared to our method, the evolution of the dynamical system is represented in the prior term in the KL divergence and the observation function in the reconstruction loss, while in our method both are represented in the reconstruction loss. The disadvantage of the method is that it does not allow to evolve the system more than one step from the estimated states, leading to reduced capacity to learn long-term dependencies.

In the experiments, we are using the same architecture for the state encoder as for the noise encoder in the DPDSR method, that is, the encoder is composed of a stack of dilated convolutional networks followed by an autoregressive LSTM network. The same parameters as for the DPDSR noise encoder are used (Tab. S1). The loss function is minimized using Adam optimizer. The optimizer is ran for 30k batch evaluations, with learning rate starting at 0.001 and reduced by a factor of 0.3 at 10k and 20k steps. The models were saved every 5k steps. The parameter sweep was performed over the observation noise variance, $\log \sigma_\eta^2 \in [-4, -2, 0]$ and initial values of the system noise variance, $\log \sigma_\epsilon^2 \in [-8, -6, -4, -2]$. As for the DPDSR method, a secondary causal encoder (Sec. A.3.1) is trained and used for the prediction tasks.

### A.3.5 AUTOREGRESSIVE LSTM (AR-LSTM)

Long short-term memory (LSTM) network (Hochreiter & Schmidhuber, 1997) is an established architecture of recurrent neural networks designed to handle long-term dependencies in the input data. Autoregressive LSTM model (Graves, 2014) represents one approach to introduce stochasticity in the standard LSTM network. The generative model is given by

$$h_t, c_t = \mathrm{LSTM}(h_{t-1}, c_{t-1}, x_{t-1}), \tag{9}$$
$$x_t \sim N(\mu_t, \sigma_t^2),$$

where $h_t$ and $c_t$ are the hidden state and cell state vectors at time $t$, and $\mathrm{LSTM}(h, c, x)$ represents the standard LSTM cell. The observation $x_t$ at each step are drawn from a normal distribution, and fed back to LSTM as an input in the next step. The parameters of the normal distribution are computed by a linear projection from the hidden state

$$[\mu_t; \log \sigma_t^2] = A h_t + b.$$

For training the model, the timeseries are split into chunks $\boldsymbol{x}$ with length $T = T_{\text{past}} + T_{\text{pred}}$. The initial conditions $[h_0; c_0; x_0]$ are first estimated from the past observation of length $x_{1:T_{\text{past}}}$. This is done using a stack of dilated convolutional neural networks, mirroring the architecture of the state encoder in our method. From the last layer and the last element in time the low-dimensional representation of the initial conditions $z_0 \in \mathbb{R}^d$ is computed via linear projection. Then the full dimension initial conditions are computed through two layer MLP with ReLU nonlinearity.

From the initial conditions the system is evolved according to (9) to obtain the simulated observations $\tilde{\boldsymbol{x}}_{T_{\text{past}}+1:T}$ and means and variances $\boldsymbol{\mu}_{T_{\text{past}}+1:T}, \boldsymbol{\sigma}_{T_{\text{past}}+1:T}$. Using the principle of scheduled sampling, during the evolution the observations entering the LSTM cell are either randomly sampled from the last step prediction ($\tilde{x}_t \sim N(\mu_t, \sigma_t^2)$) with probability $\gamma$, or replaced by the data $x_t$ with probability $1 - \gamma$.

The loss function for one sample $\boldsymbol{x}$ is given by

$$L(\boldsymbol{x}) = -\log p(\boldsymbol{x}_{T_{\text{past}}+1:T} \mid \boldsymbol{\mu}_{T_{\text{past}}+1:T}, \boldsymbol{\sigma}^2_{T_{\text{past}}+1:T}).$$

The loss function is minimized using Adam optimizer. The optimizer is ran for 30k batch evaluations, with learning rate starting at 0.001 and reduced by a factor of 0.3 at 10k and 20k steps. The models were saved every 5k steps. The parameter sweep was performed for parameter $\gamma \in [0., 0.2, 0.4, 0.6, 0.8, 1.0]$ and for prediction length $T_{\text{pred}} \in [20, 50, 100, 200]$.

### A.4 EVALUATION CRITERIA

Dynamical system reconstruction aims at training models that can robustly reproduce the temporal patterns observed in the training data on long-term scale. In this spirit we evaluate the models using two measures evaluating long-term behavior, distribution distance $D_d$ and spectral distance $D_s$, and one measure of short-term prediction capacity, 20-step prediction error $\text{PE}_{20}$. The first two are evaluated by a comparison of long model-generated time series with the original data. To generate the data, we take a point on the embedded state trajectory (via trained projection embedding method, or time delay embedding depending on the model). Using this point as initial conditions, we evolve the system for 40000 steps, using random noise for stochastic models.

The distribution distance $D_d$ measures the similarity of the distribution of the original and generated data in the observation space. To calculate it, we take the original and simulated data, collapse them across time, and compare the distributions using the Wasserstein distance (also known as Earth mover's distance). Loosely speaking, the Wasserstein distance correspond to the cost of reshaping one distribution into the other by transporting the mass. In one dimension, the Wasserstein distance between two probability distributions $u$ and $v$ with cumulative distribution functions $F_u$ and $F_v$ is defined as:

$$D_d(u, v) = \int_{-\infty}^{\infty} |F_u(x) - F_v(x)| \; dx. \tag{10}$$

We use the SciPy implementation for computations.

The spectral distance $D_s$ measures the similarity of the long time series in frequency space. To calculate it, we compute the power spectral density of the original and simulated data using Welch's method (using the SciPy implementation) with segment length equal to 4096 points. We smooth the frequency spectra using a Gaussian filter with $\sigma = 2$ time points, normalize them, and calculate their Hellinger distance. The Hellinger distance for two discrete distributions $U$ and $V$ is given by

$$D_s(u, v) = \frac{1}{\sqrt{2}} \sqrt{\sum_i \left( \sqrt{u_i} - \sqrt{v_i} \right)}.$$

We also consider a measure of short term prediction capability, the $n$-step prediction error. For data chunk $\boldsymbol{x} = (x_1, x_2, \ldots, x_T)$ we use the first $k$ time points to estimate the latent state at time $k$-th step. We then repeatedly simulate the next $n$ steps to with random noise obtain predictions $\tilde{\boldsymbol{x}} = (\tilde{x}_{k+1}, \ldots, \tilde{x}_{k+n})$. The prediction error is then

$$\text{PE}_n = \frac{1}{n} \sum_{i=1}^{n} \|x_{k+i} - \tilde{x}_{k+i}\|,$$

which we average over 20 random noise samples (for probabilistic models only) and 2000 chunks from the test dataset.

For the ECG dataset, we also measure the distance between the original and simulated distributions of the interspike intervals $D_{\mathrm{ISI}}$. To do so, use the long time series generated as described above, and we detect the peaks in the signal using the SciPy `find_peaks` tool with height 2 and prominence 1. We then then set $D_{\mathrm{ISI}}$ to be the Wasserstein distance (10) between the ISI distributions from the data and simulated signal.

We calculate the overall score as weighted sum of the distribution distance $D_d$, spectral distance $D_s$, 20-step prediction error $\mathrm{PE}_{20}$, and (for ECG) the distance of ISI distributions $D_{\mathrm{ISI}}$.

$$S = w_1 D_d + w_2 D_s + w_3 \mathrm{PE}_{20} + w_4 D_{\mathrm{ISI}}.$$

Given the different nature of the datasets, we set the weights differently across datasets. For the Lorenz and Cell datasets, we use $\boldsymbol{w} = (w_1, w_2, w_3, w_4) = (1, 1, 1, 0)$. For the Double well, RNN, and Neuron datasets, where the predictability is lower, we reduce the weight on $\mathrm{PE}_{20}$, $\boldsymbol{w} = (1, 1, 1, 0.2, 0)$. For the ECG dataset, we include the distance of interspike intervals, $D_{\mathrm{ISI}}$ as an important measure of the reconstruction quality, $\boldsymbol{w} = (1, 1, 1, 0.05)$. The weight is chosen lower due to large magnitude of unnormalized values (Fig. 4). All measures are evaluated on a test dataset which was not used for training the model.

## A.5 ANALYSIS OF THE ATTRACTORS

To identify the attractors in the state space of the model, we randomly select 100 points on the state space trajectory obtained by projecting the training data into the state space using the trained encoder. We then simulate the system forward for $T_{\mathrm{warmup}} + T$ steps with $T_{\mathrm{warmup}} = 1000$ and $T = 20000$. For stochastic models, we set the noise in the simulation to zero. After discarding the first $T_{\mathrm{warmup}}$ steps to allow the transients to decay, the remaining trajectories are analyzed to detect distinct attractors. For each trajectory we compare it against previously identified attractors by computing pairwise distances between trajectory points. Specifically, for each candidate new attractor trajectory $\boldsymbol{z}^a \in \mathbb{R}^{T \times d_z}$ and an already identified attractor $\boldsymbol{z}^b \in \mathbb{R}^{T \times d_z}$, we compute two way distances between the attractors,

$$d_{a \to b} = \mathrm{Q}_{t_a} \left( \min_{t_b} \| z_{t_a}^a - z_{t_b}^b \|, 0.8 \right),$$

$$d_{b \to a} = \mathrm{Q}_{t_b} \left( \min_{t_a} \| z_{t_a}^a - z_{t_b}^b \|, 0.8 \right),$$

where $\mathrm{Q}(\cdot, q)$ denotes the $q$-th quantile. We define the attractor distance as $d_{a,b} = \max(d_{a \to b}, d_{b \to a})$, and consider the attractors to be distinct if $d_{a,b} > \mathrm{tol}$, with $\mathrm{tol} = 10^{-5}$ if the already identified attractor $b$ is a fixed point, and $\mathrm{tol} = 10^{-1}$ for a limit cycle or a chaotic attractor. We use the quantile instead of maximum for computing the distances, and relatively high tolerances, both for more robust solution when dealing with trajectories with finite length; this is at a cost of possibly conflating close attractors.

For each new attractor we calculate the maximal Lyapunov exponent $\lambda_{\max}$ numerically (Sprott, 2003). In the algorithm, we take a point on the attractor and a point with a small perturbation. We repeatedly advance the trajectories from the initial points using the deterministic dynamics, while rescaling the deviation of the perturbed trajectory to its original norm at every step. We advance the system for 1000 time steps for, and calculate $\lambda_{\max}$ as an average from the estimates from all steps. We consider the attractor chaotic if $\lambda_{\max} > 0$, limit cycle if $\lambda_{\max} \leq 0$ and the trajectory does not converge to a single point (with $\mathrm{tol} = 10^{-5}$), and fixed point otherwise.

## A.6 USE OF LARGE LANGUAGE MODELS

Large Language Models (LLMs) were used in preparing the manuscript to polish the writing. LLMs were also used, together with specialized literature search tools, to identify relevant research works.

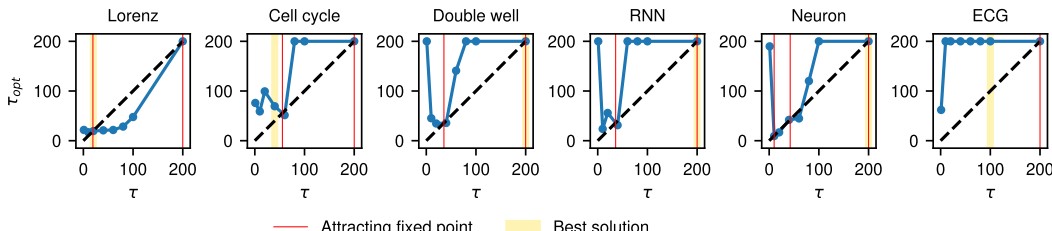

Figure S1: Optimal value of teacher forcing interval $\tau_{\mathrm{opt}}$ across datasets. For each value of $\tau$, we calculated the maximal Lyapunov exponent $\lambda_{\max}$ along the estimated trajectory using the final trained model. Following Mikhaeil et al. (2022), we calculate $\tau_{\mathrm{opt}} = \log 2/\lambda_{\max}$ for $\lambda_{\max} > 0$, and we set $\tau_{\mathrm{opt}}$ equal to the maximal trajectory length (=200) otherwise. The red lines indicate the attracting fixed points of a relaxed fixed point iteration scheme $\tau_{i+1} = (1 - \alpha)\tau_i + \alpha\tau_{\mathrm{opt},i}$ if ran on the visualized relation $\tau_{\mathrm{opt}}(\tau)$ obtained from the final models, that is, they show the points of diagonal crossings with derivative smaller than 1. We assume linear interpolation between the evaluated points. The yellow line indicates the position of the best trained models, same as in Fig. S6. Note that here the Lyapunov exponents are evaluated along the estimated trajectories, and not for the deterministic attractors as in Fig. S6. For most datasets, two fixed points exist, one in the low $\tau$ (deterministic) regime, and one in the high $\tau$ (noise-driven) regime, indicating that the adaptive $\tau$ scheme might not robustly converge to the optimal solution.

# B  SUPPLEMENTARY RESULTS

## B.1  CHOICE OF THE TEACHER FORCING INTERVAL

An important question for practical applications of the method is how to select the teacher forcing interval $\tau$. In this work, we have performed a parameter sweep across a range of values. Such approach, while robust, is costly in terms of computational time. Existing works proposed several approaches to similar issues in deterministic models. Mikhaeil et al. (2022) used teacher forcing interval equal to the predictability time of a chaotic system

$$\tau_{\mathrm{opt}} = \frac{\log 2}{\lambda_{\max}} \tag{11}$$

with $\lambda_{\max}$ being the maximal Lyapunov exponent estimated ahead from the data; they showed that such estimates match closely the optimal values. Hess et al. (2023) introduced an adaptive scheme where the parameter of generalized teacher forcing is updated during the training based on the product of Jacobians of the trained system, evaluated along the forced trajectory.

In the first approach, estimating the Lyapunov exponent from the data is performed under the assumption of deterministic chaotic system, and therefore is unsuitable for our purposes. However, a possible option would be to combine the two methods, that is, an adaptive scheme where at each step $i$ of the training we estimate the maximal Lyapunov exponent along the forced trajectory of our stochastic system, calculate the optimal interval $\tau_{\mathrm{opt},i}$ using (11), and adapt the parameter following a relaxed fixed point iteration scheme

$$\tau_{i+1} = (1 - \alpha)\tau_i + \alpha\tau_{\mathrm{opt},i} \tag{12}$$

with relaxation parameter $\alpha \in (0, 1]$.

For a preliminary investigation if such approach is feasible, we reanalyze the results from our parameter sweep, and calculate $\tau_{\mathrm{opt}}$ from the final trained models for each constant value of $\tau$ (Fig. S1). Using this visualization we can see where the equation $\tau_{\mathrm{opt}}(\tau) = \tau$ solved by scheme (12) has its attracting fixed points. These can provide an indication to which values of $\tau$ might the hypothetical adaptive scheme converge. While it is not guaranteed that the adaptive scheme would converge to the same solutions as the scheme with fixed $\tau$, these results suggest that the scheme might not be sufficiently robust. On most datasets we see the existence of two attracting fixed points: one in the low range of $\tau$ in the deterministic regime with chaotic dynamics, and one in the noise-driven regime, where the negative Lyapunov exponent results in the maximal teacher forcing interval. Although

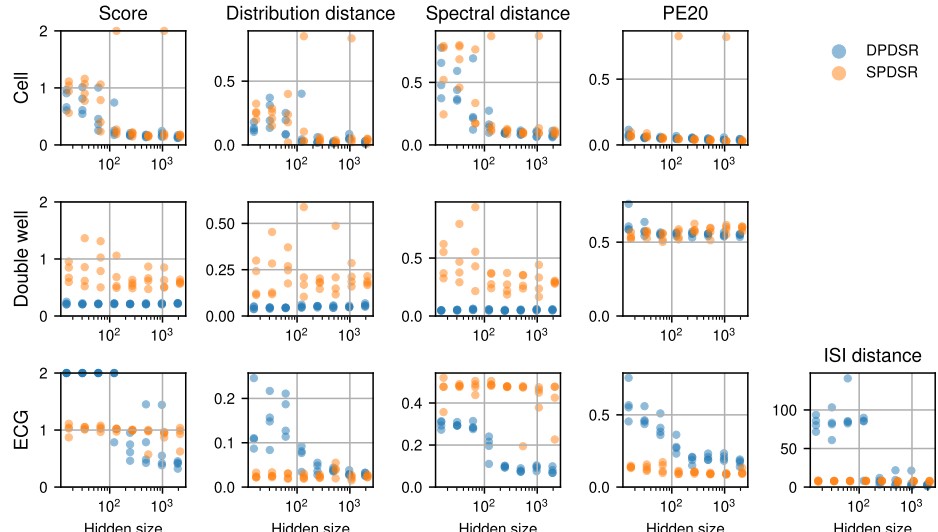

Figure S2: Effect of the number of units in the hidden layers. The effect on the model quality was investigated for three datasets (rows), the score and the separate measures of reconstruction quality are in the columns. For clearer visualization, the results were clipped at the upper limit of the shown range.

the best solution closely matches one of these fixed points for all but one problem, the adaptive scheme might not be able to correctly identify the optimal one. We therefore conclude that for practical purposes where robust results are required the parameter sweep remains the best option, and we highlight of task of finding the optimal teacher forcing strategy in stochastic approaches as an interesting research problem for future studies.

## B.2 ARCHITECTURE VARIATIONS

### B.2.1 NUMBER OF PARAMETERS IN THE GENERATIVE MODEL

We investigated how does the performance of the stochastic DPDSR method change when the number of parameters in the generative model is increased or decreased. In particular, we aimed at a comparison with the behavior of its deterministic modification: the SPDSR method. We considered three problems, the Cell, Double well, and ECG datasets. For each, we trained stochastic and deterministic models with 16 to 2048 units in the hidden layer of the generative model (2), and evaluated the models as before.

Fig. S2 shows that the behavior of the method depends on the dataset. For the Cell dataset, generated by a low-dimensional deterministic model, both DPDSR and SPDSR behave similarly, with increasing performance for increasing number of parameters. For the Double well dataset, generated by a noise-driven model, the stochastic model performs well even for minimal number of parameters, and consistently outperforms the deterministic model. For the ECG dataset we see that the deterministic models perform equally well across the range of parameters. The stochastic models, however, perform considerably worse for models with less than 100 units in the hidden layer, but outperform the deterministic models for larger hidden layers.

### B.2.2 ENCODER ARCHITECTURE

We further investigated the role of the encoder architecture on the performance of the DPDSR method (Fig. S3). Specifically, we compared the baseline architecture with two modifications: First, a variation with a noise encoder with the autoregressive RNN removed, and with the parameters otherwise kept equal as described in Tab. S1. Second, a variation with a autogressive RNN in noise encoder also removed, but with increased number of channels (26 instead of 24) in the dilated convolutional neural network to keep the total number of parameters approximately equal. Degradation

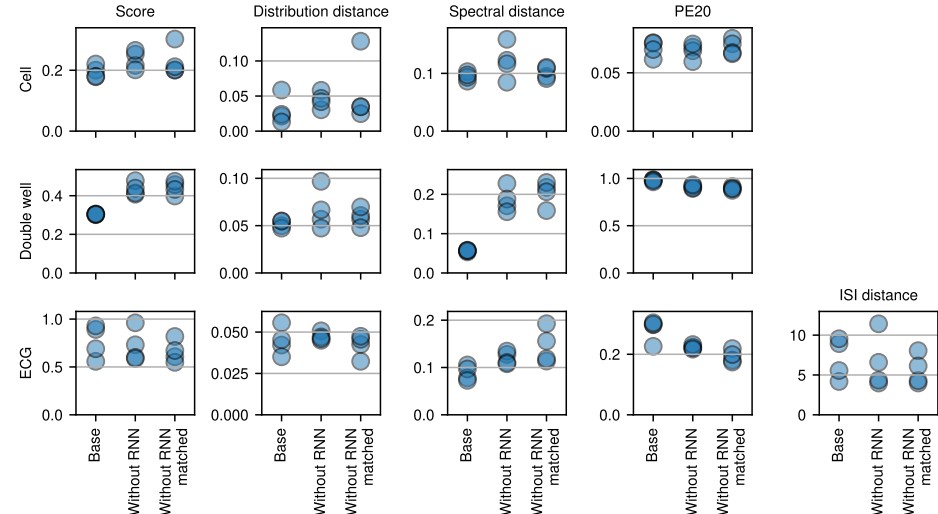

Figure S3: Role of the encoder architecture. For three datasets, three variations of the DPDSR model architecture were investigated: the baseline, a variation with the autoregressive RNN in the noise encoder removed, and a variation with the autoregressive RNN removed, but increased number of parameters in the convolutional neural network to match the total parameters. For each variant, four models with different random weight initializations were trained.

in performance with the removed RNN can be seen for the Double well problem, while for the Cell and ECG problems the modification have little impact.

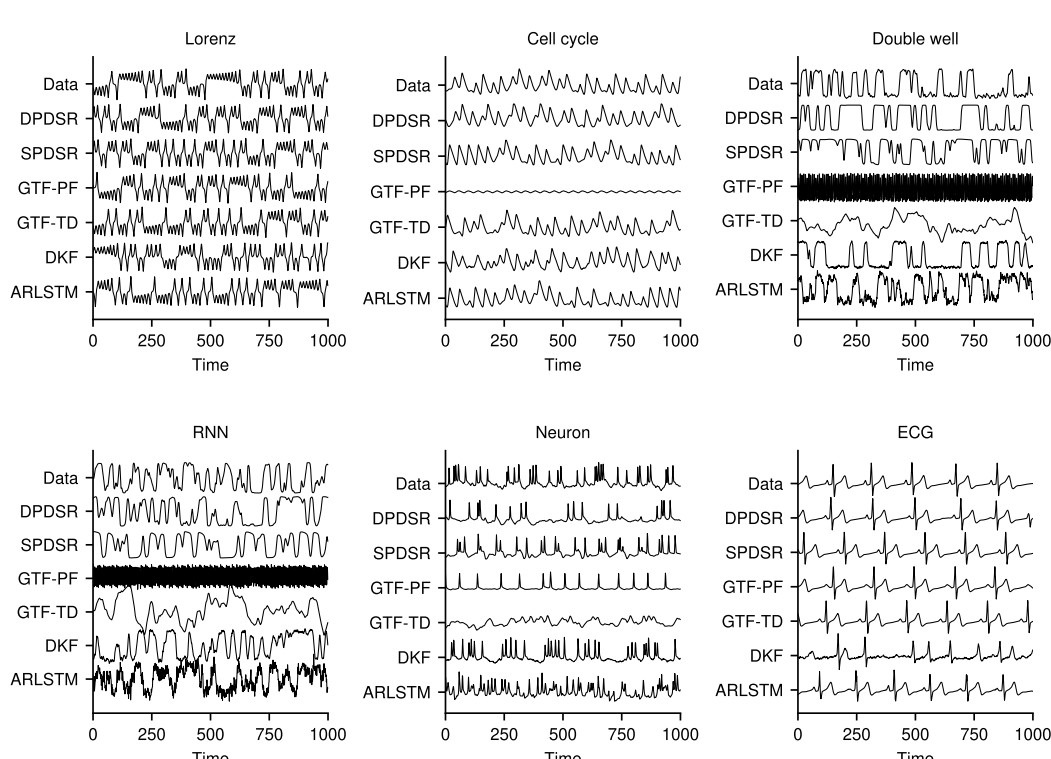

Figure S4: Examples of generated time series for all test problems and evaluated methods. The time is represented in sample indices and not the original model or real time.

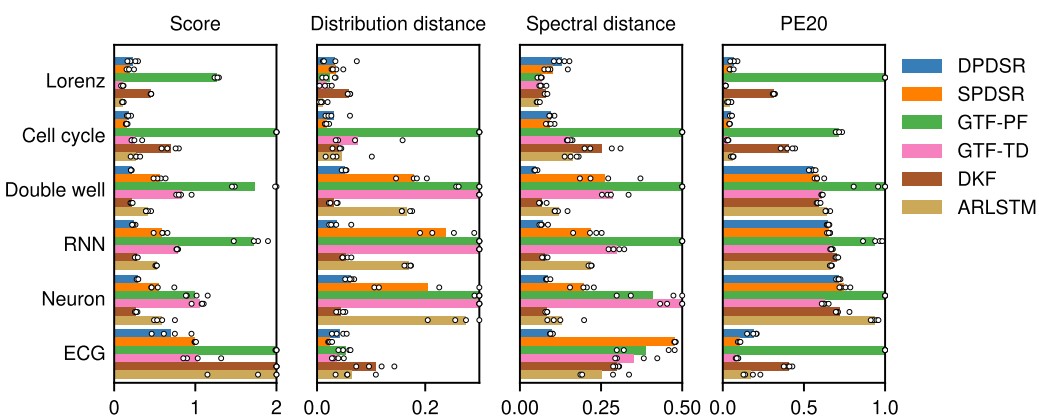

Figure S5: Reconstruction quality of the evaluated methods for the six datasets. The panels show the overall score, distribution distance $D_d$, spectral distance $D_s$, and the 20-step prediction error $\text{PE}_{20}$ (lower = better for all measures). Each circle represent one of four initializations of the training method, and the bar shows the mean. For clearer visualization, the results were clipped to the shown interval.

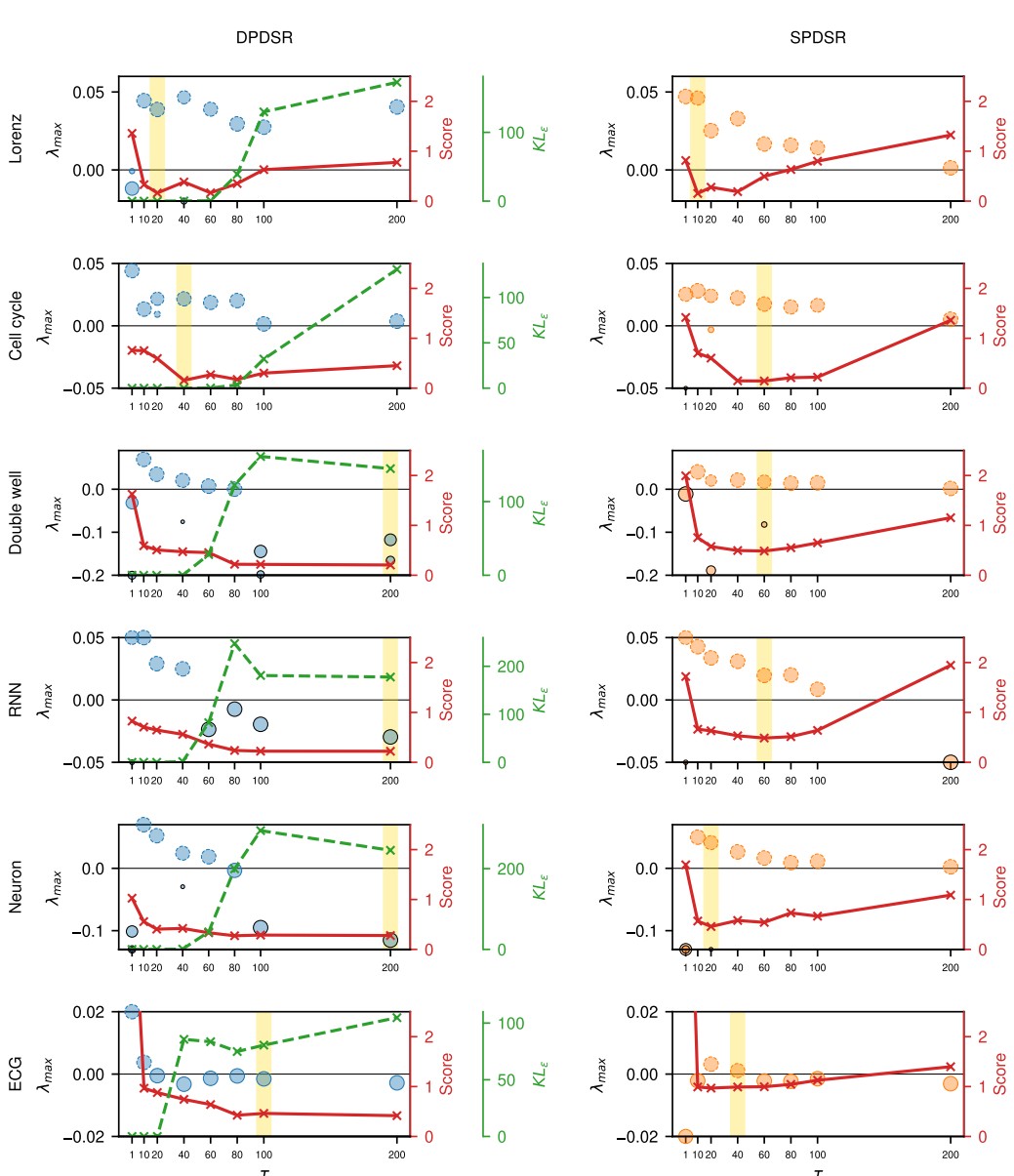

Figure S6: Analysis of the attractors of the trained models for all datasets, extending Fig. 5. Left column: DPDSR models (stochastic); right column: SPDSR models (deterministic). Circles show the maximal Lyapunov exponent. Dashed circle outline represents a chaotic attractor ($\lambda_{\max} > 0$), colored solid outline represents a limit cycle, and black solid outline a fixed point. Size of the circle corresponds to the size of the basin of attraction. Solid red line shows the score (lower = better). Dashed green line shows the KL divergence of the estimated posterior distribution of the noise to the prior distribution $\mathrm{KL}_\epsilon = D_{\mathrm{KL}}(q(\epsilon \mid x) \parallel p(\epsilon))$. The yellow band indicates the optimal $\tau$ value for the dataset; note that the optimal $\tau$ value is selected based on the mean score across all initializations, while for other visualizations we plot only the best model. For clearer visualization, the positions of the Lyapunov exponents are clipped to the shown interval.