# OpenReview forum: "Dynamical system reconstruction from partial observations using stochastic dynamics"
_ICLR.cc/2026/Conference — ICLR 2026 Conference Withdrawn Submission_

### Official Review · Reviewer_MPg2 · 2025-10-25

**Soundness:** 1
**Presentation:** 1
**Contribution:** 1
**Rating:** 0
**Confidence:** 4

**Summary:**

The authors propose a training method called DPDSR (Double Projection Dynamical System Reconstruction). The framework focuses on a generative model where, at each time step, the input consists of the previous latent state (deterministic) and stochastic noise (probabilistic). The observation model is implemented as a simple MLP that takes the latent state and adds observation noise.

The key contribution lies in motivating a new training algorithm, with the loss inspired by the standard ELBO formulation but modified in three main ways:

The noise is modeled probabilistically, while the latent state remains a point estimate.

The prior model is applied for $\tau$ steps before being replaced by the estimated latent state.

An additional regularization term is introduced.

**Strengths:**

Revisiting the classical ELBO formulation in the context of latent variable models is of broad interest to the community.

The attempt to explore a hybrid treatment of latent states (deterministic) and noise (probabilistic) is an interesting angle.

**Weaknesses:**

Overall, the paper introduces a series of heuristics that deviate from a principled ELBO-based training framework. These raise concerns about both theoretical soundness and novelty:

The use of teacher forcing for $\tau$ steps does not correspond to the correct posterior distribution.

It is unclear how Equations (6) and (7) can be formally derived from the original ELBO formulation.

The claimed novelty is limited: estimating noise instead of latent states has been studied in a principled way in prior work (e.g., Mind the Gap When Conditioning Amortized Inference in Sequential Models).

The experimental evaluation is weak and raises questions about reproducibility and fairness.

**Questions:**

Without a proper theoretical foundation, I cannot recommend acceptance. To strengthen the paper, I suggest the following clarifications and additions:

Can you provide a rigorous theoretical discussion of the training algorithm, motivating each design choice? Ablation studies would also help disentangle the contributions of individual components.

Can you include detailed and reproducible evaluation metrics instead of aggregate scores that cannot be verified? Benchmarking against established datasets would also make the results more credible.

Can you demonstrate your method using more flexible and competitive architectures? At present, restricting the analysis to very simple MLPs undermines the applicability and strength of your claims.

---

### Official Review · Reviewer_mmdr · 2025-10-27

**Soundness:** 2
**Presentation:** 2
**Contribution:** 2
**Rating:** 2
**Confidence:** 3

**Summary:**

The authors propose a method to learn dynamics of stochastic systems. They propose to use an encoder structure that infers both a state variable z and noise variables epsilon. The authors claim that their proposed encoder is able to infer partially observable dynamics. To improve training, they use a teacher-forcing strategy.

**Strengths:**

-	I think its an interesting idea to estimate the noise with an encoder, and make it fit to a standard gaussian prior.
-	I think your approach of evaluating your training results as described in section 4 is promising.

**Weaknesses:**

-	Differentiation from other methods is unclear, and why this approach should be better. The authors do   not clearly asses which limitations of VAE based frameworks they addres with the proposed methods.
-	It would make the paper better if the research gap you are trying to fill would be better described. For example, you write in Line 59 “But these approaches were not yet sufficiently explored, in particular their performance with algorithms used for training stochastic dynamical systems.” Please be more specific. Is there any work on partially observed systems? Can you improve your literature review and exactly define the research gap you are trying to fill? In return, you could try to shortend line 24- 56. Also, in line 15 in the abstract you talk about limitations, it would be good, to be specific.
-	Not clear what the authors mean by observed and unobserved variables exactly, i.e. how they are included in the framework, and which datasets have which unobserved variables. It would be good to describe the differences between observations x and states z and how this differs in the dataset. It seems like that the states z describe actually the states of the dynamic system and that a ground truth for them (partially) exists. However, how do you choose which variables of a dataset are states?
-	In contrast to the also mentioned continuous-time methods, the proposed method is depended on a regualarily spaced time-scale.
-	They say, that they evaluated noise is only on one variable, making the method not that performant?
-	Is Line 28: “Compared with the related task of time series prediction, the problem of dynamical system reconstruction (DSR) is distinguished by the following aspects: Compared with the related task of time series prediction, the problem of dynamical system reconstruction (DSR) is distinguished by the following aspects:” a definition by you, or can it be referenced by some sources? This would make the paper stronger as you often refer to it.
-	I think in line 77 ongoing, you talk about Dynamical Variational Autoencoders. You should name them after the Kingma citation, and refer to Girin’s review.
-	In line 127ff., you talk about “discretize-then-differentiate” and continuous-time models. As you don’t employ a continuous-time model, this is not really necessary. Additionally, “discretize-then-differentiate” is usually named “discretize-then-optimize” and a citation for this, e.g. the PhD thesis of Kidger, should be given.
-	Description of the model (section 3.1): hard to understand. What do authors mean by observed an unobserved variables? Where is the identy function the talk about in equations 5-7? Why do you name only epsilon a latent variable? I think, z would also be latent variables, as they are inferred.
-	In you description of your model you follow the structure proposed in Girin et. Al. Please cite that.
-	In line 180 you say that you only use one-dimensional noise in the last dimension in your presented examples. To me, this seems is the method is not really performant in predicting stochastic systems. Can you say anything against that, or why you did this?
-	In your motivation for you new methods in line 151-160:
-	I would disagree that Dynamical Variational Autoencoders only make 1-time-step predictions (around line 153). It just depends on how you specify the exact training strategy.
-	In line 158, you claim that teacher-forcing strategies are “crurical” for training deterministic systems. I would disagree, as we do it in our research without it. Can you a) explain better, what teacher forcing does, and b) why you see so much promise in employing this?
-	The motivation would better be used in the introduction or review of related work.
-	Compared methods (3.2): Why do you not test more methods of the review of Girin et al, only DKF? I don’t understand how a standard LSTM has a probabilistic output?  It would be good to provide a reference where you describe in more detail the used architectures.
-	Used datasets: No citations for datasets 3-6. It would be good to know which other studies used this dataset as well.
-	The authors say that they adapt the weights for the overall score depending on the dataset. This raises the question, if weights are adjusted to achieve that the proposed methodology achieves the best performance, making it hard for the reader to evaluate the performance properly.
-	It would be good to describe what infromation the maximal Lyapunov exponent describes and which numerical values describe which behaviour, as it is hard to understand the following evaluation.
-	It seems like the discussion paragraph wasn’t finished. Lines 463-485 reads like a literature review and its contents might be better suited to the related works section.
-	A conclusion is missing.

Minor:
-	Line 77 no “the” before data
-	Line 53, what do you mean by “autonomous”?
-	Line 107, what is “Euler-Maruyama” discretization? Isn’t it just  Euler?
-	Explanation of the abbreviation PECUZAL is missing (line 242)

**Questions:**

-	Is teacher forcing comparable to multiple-shooting in Neural ODEs? Would it be possible to employ growing horizon techniques, e.g. start by predicting short sequence length and slowly increasing the prediction lengths?
-	Why is the double projection approach necessary and do other publications do the same?

If the authors address my concerns, especially by better explaining the need for this work, its setting in the related literature, improve the presentation of the method itself, and add a proper conclusion, I’m willing to increase my score.

---

### Official Review · Reviewer_qXcw · 2025-10-28

**Soundness:** 3
**Presentation:** 4
**Contribution:** 4
**Rating:** 4
**Confidence:** 3

**Summary:**

This paper proposes DPDSR (Double Projection Dynamical System Reconstruction), a VAE-based method for learning stochastic dynamical systems from partially observed time series. The key idea is to encode observations into both estimated state trajectories and noise time series, then use teacher forcing on states while treating noise as latent variables. The authors test on six problems, including synthetic chaos (Lorenz, cell cycle), noise-driven systems (double well, RNN), and real data (neuron voltage, ECG). Results show the method handles stochastic systems better than deterministic alternatives, though with high variance across initialisations.
The core contribution is reasonable: existing methods either can't do multi-step evolution (Deep Kalman Filter) or can't use teacher forcing (STORN). This dual-encoding approach aims to get the best of both worlds.

**Strengths:**

- Using two separate encoders—one for states (enabling teacher forcing) and one for noise (enabling proper stochastic modeling)—is a neat solution to a genuine problem. The motivation is clear, and the approach makes intuitive sense.

- The dynamical analysis in Fig. 5 and S6 is genuinely insightful. Showing how the teacher forcing interval τ controls whether you get deterministic chaos vs. noise-driven dynamics is interesting.

- Six test problems with different characteristics (deterministic chaos, noise-driven, experimental) are a decent scope. The ECG analysis revealing "skipped beats" as a failure mode is the kind of honest reporting that's useful.

- Including both stochastic (DKF, AR-LSTM) and deterministic (GTF, SPDSR) baselines makes sense for understanding where stochastic modelling helps

**Weaknesses:**

- Four random initialisations are not enough, period. Look at your own Fig. 3—the variance is huge. Some methods have error bars spanning the entire range. You're making claims about which method is "better" based on the means of 4 samples with massive variance. Same with Figure 4D.

- You're setting different weights (w1, w2, w3, w4) in the score function for each dataset. Why? The weights seem arbitrary and thus could arbitrarily favour your method.

- Why WaveNet? We need an additional ablation study to demonstrate the method's value, or switch to a more standard model for those tasks.

**Questions:**

- You claim PECUZAL "fails" on stochastic systems, but the original paper explicitly says it handles noisy time series. Why the discrepancy?

- Tanh is applied on both the diffusion term and the drift (eq. 1). This doesn't sound standard, as tanh on the noise does change its property. Is that what you are doing?

- DKF performs terribly on everything, including problems where it should work reasonably well. Can you comment on that?

- Can you increase statistical power?  Even if computational cost is high, you need more than four initialisations for the final version.

- How sensitive are results to the score weights? Show what happens when you use uniform weights (1,1,1,0) across all datasets.

---

### Official Review · Reviewer_M7Jh · 2025-10-29

**Soundness:** 3
**Presentation:** 3
**Contribution:** 3
**Rating:** 6
**Confidence:** 2

**Summary:**

This paper introduces a novel method for Dynamical System Reconstruction (DSR) from partially observed data, named Double Projection Dynamical System Reconstruction (DPDSR). The core idea is to use a variational autoencoder (VAE) framework to jointly estimate both the latent system state trajectories and the stochastic driving noise time series from the observed data. A key feature is the integration of a teacher forcing strategy, where the simulated state is periodically reset to the encoder's state estimate, which helps in training the stochastic dynamics over long horizons. The paper demonstrates that DPDSR performs competitively across all problem types, whereas existing methods (deterministic or stochastic) excel only on specific subsets. A further analysis investigates how the teacher forcing interval influences the nature of the learned dynamics, revealing a transition from deterministic-chaotic regimes (low $\tau$) to noise-driven regimes (high $\tau$).

**Strengths:**

* The "double projection" concept is a novel solution to a known limitation in stochastic VAE-based DSR. By encoding both states and noise, it bridges the gap between methods that use states as latents and those that use noise as latents.

* The experimental evaluation is extensive and robust. The use of many distinct datasets, the fair and careful matching of model parameters, the multi-criteria evaluation score, and the analysis of the learned attractors' properties all lead to a high-quality empirical study.

* One of the paper's greatest strengths is that it does not stop at reporting error metrics. The analysis in Figure 5, which uses Lyapunov exponents to probe the internal dynamics of the learned models, provides a deep and significant scientific insight: deterministic models "fake" noise with chaos, while the proposed stochastic model can learn the true, simpler, noise-driven dynamics .

* The paper provides strong evidence that DPDSR is a versatile and robust DSR method. The analysis of the $\tau$-dependent regime shift (chaotic vs. noise-driven) offers valuable insights into the training dynamics.

**Weaknesses:**

* The method's performance and fundamentally the nature of its learned solution (chaotic vs. noise-driven) are critically dependent on the teacher forcing interval ($\tau$). The authors correctly identify this as a limitation, as it requires an expensive hyperparameter sweep.

* The authors note that the DPDSR model for the ECG data can "skip a beat" and that this behavior is sensitive to initialization. This suggests the learned attractor, while good, may have minor instabilities or undesired properties, which is a slight weakness in the method's robustness. However, again this was correctly identified and acknowledged by the authors.

**Questions:**

* The authors are transparent about the "skipped beat" phenomenon in the ECG model. Have you investigated the cause of this? Does it correspond to a specific, unstable region of the learned limit cycle, or a particular sequence of inferred noise that pushes the trajectory off-attractor? Does this issue persist with different noise regularizations?

* About the analysis of the $\tau$-dependent regimes: does the DPDSR model with a very small $\tau$ (e.g., $\tau=1$) and its correspondingly low noise KL-divergence (Fig 5) effectively "shut off" the stochastic component and converge to the same solution as the deterministic SPDSR model?

---

### Official Review · Reviewer_5GmW · 2025-10-31

**Soundness:** 3
**Presentation:** 3
**Contribution:** 3
**Rating:** 6
**Confidence:** 4

**Summary:**

The paper introduces Double Projection Dynamical System Reconstruction (DPDSR), a stochastic variational framework for learning latent dynamical systems from partial and noisy observations. Unlike standard VAE-based state-space models (e.g., Deep Kalman Filters), DPDSR performs two parallel projections:

- An encoder that infers latent system states $𝑧_𝑡$.
- A second encoder that infers latent driving noise $\epsilon_t$.
The method is trained end-to-end using teacher forcing over multi-step rollouts, allowing it to balance deterministic and stochastic dynamics.

A systematic evaluation is carried out on six datasets—three synthetic (Lorenz, Cell Cycle, Double-Well) and three empirical (RNN, Neuron, ECG)—and compared against deterministic reconstruction (SPDSR, Generalized Teacher Forcing), stochastic baselines (Deep Kalman Filter), and AR-LSTM. The model consistently performs competitively or best across all datasets, with detailed dynamical analysis of the learned attractors (Lyapunov spectra, noise-driven vs. chaotic regimes).

**Strengths:**

- The double-projection idea (state + noise) is original within the DSR context. It disentangles deterministic evolution from stochastic excitation, which aids interpretability of noise-driven dynamics.

- By varying the teacher-forcing interval, the method interpolates between deterministic chaos and noise-driven behavior. This dual regime analysis provides useful insight into the transition between chaotic and stochastic attractors.

- Six diverse datasets, including empirical ones, make the benchmarking solid. The multi-criteria score (distribution, spectral, prediction errors) captures both long-term and short-term fidelity.

- Estimation of Lyapunov exponents and attractor structures is uncommon in ML papers and adds insight.

- The combination of autoregressive posterior, WaveNet-based encoders, and teacher forcing improves stability and expressivity, especially for partially observed systems.

**Weaknesses:**

- Despite the “double projection” framing, DPDSR remains close to stochastic latent-state VAEs (DMM, SRNN, VRNN). The argument that DKF/DMM cannot capture long-term dependencies because they are trained with one-step losses is not fully convincing; multi-step or rollout training is straightforward to integrate into DMM-style frameworks (and is usually avoided in favor of near real time frameworks).

- Key related baselines such as Deep Markov Models, Latent SDEs, or Neural ODE variants are discussed but not empirically compared. This omission weakens claims of general superiority.

- Model performance seems sensitive to the forcing interval $\tau$. The paper acknowledges this but offers no principled criterion or adaptive strategy; tuning τ by sweep undermines the method’s autonomy.

- Dual encoders, autoregressive posteriors, and repeated multi-step rollouts increase training cost substantially relative to DKF or DMM, without a clear demonstration of efficiency gains.

- Despite the motivation of reconstructing interpretable dynamics, the fully connected neural parameterization offers little structural insight beyond what DMMs already provide. Scalability to high-dimensional continuous systems is not addressed.

- The method is presented as a distinct stochastic reconstruction framework, yet in practice it is a specific instance of a variational state-space model with multi-step reconstruction loss.

**Questions:**

- It would help to elaborate on how the DPDSR differ mathematically from the Deep Markov Model (DMM) or Stochastic RNN (SRNN)?
- Could a DMM trained with multi-step rollout and teacher forcing achieve equivalent performance?
- Why is the double-projection (state + noise) necessary if the latent state in DMM can already encode both deterministic and stochastic components?
- How interpretable are the learned “noise” variables? Do they correspond to actual process disturbances, or are they simply a learned residual? Is there any quantification (e.g., mutual information) showing that $\epsilon_t$ encodes meaningful stochastic forcing?

- Since both $𝑧_𝑡$ and  $\epsilon_t$ are learned from data, is the decomposition unique? Could their roles be swapped or entangled during training?
- Are there constraints ensuring identifiability (e.g., orthogonality, disentanglement penalties)?

- Can the model generalize to unseen rollout lengths?

- As learning the noise is an important part of this work, I would expect a more systematic examination of performance on various levels of additive measurement noise or non-Gaussian corruptions?

- How does performance scale with increasing system dimension $𝑑_𝑧$.

---

### Note · Authors · 2025-12-02

**Comment:**

We thank the reviewers for their time and comments.

**Withdrawal Confirmation:**

I have read and agree with the venue's withdrawal policy on behalf of myself and my co-authors.